# Modified Orbitozygomatic Craniotomy Approach for a Recurrent Orbital Tumor in a Pediatric Patient

**DOI:** 10.3390/medicina60081267

**Published:** 2024-08-06

**Authors:** Róbert Chrenko, Beáta Bušányová, Anton Gerinec, Dana Tomčíková, Boris Rýchly, Marek Grega, Martin Hanko, Barbora Nedomová

**Affiliations:** 1Department of Pediatric Neurosurgery, National Institute of Children’s Diseases, Limbová 1, 833 40 Bratislava, Slovakia; robertchrenko@gmail.com (R.C.); marek.grega1@gmail.com (M.G.);; 2Faculty of Medicine, Slovak Medical University, Limbová 12, 833 03 Bratislava, Slovakia; nedomova76@gmail.com; 3Department of Pediatric Ophthalmology, National Institute of Children’s Diseases, Limbová 1, 833 40 Bratislava, Slovakia; gerinec@azet.sk (A.G.); dtomcikova@gmail.com (D.T.); 4Faculty of Medicine, Comenius University, Špitálska 24, 813 72 Bratislava, Slovakia; 5Department of Pathology, Unilabs Slovensko, s.r.o., Polianky 7, 841 01 Bratislava, Slovakia; boris.rychly@unilabs.sk; 6Bory Hospital—Penta Hospitals, Ivana Kadlečíka 2, 841 03 Bratislava, Slovakia; 7Department of Pediatric Anesthesiology and Intensive Medicine, National Institute of Children’s Diseases, Limbová 1, 833 40 Bratislava, Slovakia

**Keywords:** modified orbitozygomatic craniotomy, orbit, pediatric, tumor

## Abstract

*Background*: This report aims to present the case of a pediatric patient with a recurrent tumor in the superolateral orbit. *Clinical Presentation*: An 8-year-old patient was initially treated for a tumor in the superolateral orbit via a transconjunctival approach. The histopathological diagnosis was epidermoid cyst. Postoperatively, chronic inflammation and fistula developed in the lateral canthus area. Magnetic resonance imaging revealed a residual tumor posterior to the original tumor location. The patient was treated via a modified orbitozygomatic (mOZ) craniotomy approach that was originally applied in neurosurgery, and complete tumor removal was achieved. A temporary paralysis of the frontotemporal branch of the facial nerve was observed and fully resolved within one month following surgery. At the 18th month of follow-up, the visual, neurological, and cosmetic results were found to be satisfactory. *Conclusions*: mOZ craniotomy can be used to access and operate on recurrent orbital tumors in pediatric patients where other more aggressive surgical approaches should be avoided.

## 1. Introduction

The orbital compartment is affected by various pathological lesions. Orbital pathological conditions encompass a heterogeneous group of pathological entities, including neoplastic and non-neoplastic lesions. The incidence of primary orbital tumors is low, affecting approximately 1 in 100,000 people [1]. Dermoid cysts account for about 3–9% of all orbital masses, and they are one of the most common orbital tumors in children, accounting for more than 40% of childhood orbital lesions [2]. Intraorbital epidermoids are even more infrequent. In the upper outer orbital quadrant, dermoid cysts have been the most prevalent type of tumor identified [3]. These lesions have a wide variety of clinical and radiological presentations, and they should be considered in the differential diagnosis of orbital cystic lesions. They usually warrant surgical intervention and should be completely excised because of the risk of recurrence or inflammation and infection risks due to their retained contents. They tend to recur with lipogranuloma formation, and there is a remote possibility of malignant transformation [4].

Surgical approaches to orbital lesions can be complex and may require cooperation between neurosurgeons, ophthalmologists, and otolaryngologists. Considering the morbidity related to complex orbital procedures, defining the surgical strategy and the roles of participating specialists is crucial. The optimal surgical strategy is determined by the (1) location of the lesion in orbit, (2) lesion characteristics (consistency, infiltration, blood supply, extent, etc.), and (3) surgeon’s expertise [5]. Lesions in the superolateral orbit extending to the orbital apex can be operated on via a supraorbital craniotomy or a fronto-temporal craniotomy with zygomatic osteotomy of varying extent [6,7]. It is important to note that typical neurosurgical approaches are not without their share of complications. The most significant complications include blindness resulting from optic nerve injury or compromise of the central retinal artery; diplopia due to oculomotor dysfunction caused by intraocular muscular injury or damage to one or more nerves supplying the extraocular muscles; the Bernard–Horner syndrome resulting from injury to the ciliary ganglion and the sympathetic nerves, enophthalmos, pulsatile exophthalmos, brain injury, seizures, and neurological deficits due to brain injury [6].

However, the relative complexity of these approaches led us to seek a potentially less invasive approach to this region. Moreover, the young age of our patient presented additional challenges.

The surgical technique of orbitozygomatic craniotomy reported by Balasingam et al. is an excellent procedure that allows wide surgical exposure, easy orbital reconstruction, and a satisfactory postoperative esthetic outcome. The surgical technique involves a one-piece osteoplastic bone flap incorporating the frontal, temporal, and lateral portions of the orbital rim as a technically simpler alternative to the standard orbitozygomatic (OZ) craniotomy. The orbital rim component extends just laterally from the supraorbital foramen/notch to the frontozygomatic suture. This craniotomy eliminates the need to remove the zygoma. The osteoplastic bone flap minimally obstructs the surgical view and offers all the advantages of a standard OZ craniotomy. Temporalis muscle atrophy leading to temporal hollowing is avoided, bone union to the calvaria is improved, and the possibility of bone infection is reduced [8].

The aim of this article was to report a case of a recurrent superolateral orbital tumor that was successfully managed via a modified orbitozygomatic (mOZ) craniotomy approach (Figure 1). This is the first report of a recurrent orbital tumor accessed via an mOZ craniotomy in a pediatric patient.

## 2. Clinical Presentation

The patient consented to the procedure and to the publication of their image through the family. An 8-year-old patient presented with a widened right eyelash slit, 3 mm proptosis, limited abduction, and a central visual acuity of 1.0. Magnetic resonance imaging (MRI) revealed a cystic tumor measuring 20 mm in diameter localized to the extraocular superolateral space of the right orbit that exerted pressure on the lacrimal gland and rectus lateralis muscle. The lesion was extirpated using a transconjunctival approach, and bleomycin was injected into the residual lesion at the apex of the orbit. Histopathological examination confirmed the diagnosis of an epidermoid cyst of the orbit.

Postoperatively, a fistula and intermittent inflammatory symptoms developed in the orbital area and skin around the right outer canthus (Figure 1A,B).

MRI revealed a 7 mm diameter residual epidermoid cyst located posterior to the initial lesion in the superolateral extraconal area that extended to the apex of the orbit (Figure 2A–L).

Reoperation via the transconjunctival approach was considered infeasible due to the posterior location of the lesion. Second-look surgery using a transcranial approach was indicated, and an mOZ craniotomy approach was selected to access the lesion (Figure 3).

Antibiotic prophylaxis with cefotaxime (30 mg/kg) was administered intravenously 30 min before skin incision. A lumbar tap was performed, and 20 mL of cerebrospinal fluid was withdrawn. Moreover, 0.5 g/kg of mannitol was administered IV to relax the brain and allow minimal retraction of the brain and bulbus. Optic neuronavigation Medtronic Stealthstation S8 using a fusion technique involving computed tomography and MRI was performed intraoperatively. Guided mOZ craniotomy, including bone work, as schematically demonstrated in Figure 4, was performed as previously described [7].

Intraoperative photographs show the exposed operative field after mOZ craniotomy and one-piece bone flap elevation (Figure 3A,B). After periorbital incision, the tumor was removed along with its capsule using conventional microsurgical techniques (Figure 3C–F). No tumor fluorescence was detected using a 570 nm microscopic light after the administration of fluorescein (5 mg/kg IV). The periorbita was sutured using a nonresorbable monofilament suture (4–0 polypropylene) and covered with a conventional sealant. The craniotomy flap was fixed to the craniotomy border using resorbable sutures (2–0 polyglactin) via microdrill holes. The fascia of the temporalis muscle, galea, and subcutis were sutured using resorbable material, and the skin was sutured using running monofilament nonresorbable sutures. The operative time (incision–suture) was 170 min.

Complete tumor removal was confirmed on postoperative MRI (Figure 2M–O). The progressive remission of convergent strabismus was observed, and periorbital edema resolved on postoperative day (POD) 8. Transitional headache after full mobilization on POD 3 was managed conservatively (bed rest, analgesics, and infusion) and resolved completely on POD 8. The skin sutures were removed on POD 8. Temporary paralysis of the fronto-temporal branch of the facial nerve (Grade 3 on the House and Brackman scale) [9] was noted after the resolution of the periorbital edema and was managed conservatively (electrotherapy and vitaminotherapy) until it resolved completely (Grade 1 on the House and Brackman scale) [9] at 1 month postoperatively. Histopathological examination confirmed the diagnosis of epidermoid cyst (Figure 5). Satisfactory visual, neurological, and cosmetic outcomes were observed during ophthalmological and neurosurgical outpatient examinations at 1 month postoperatively (Figure 1C). The patient was followed-up with by a pediatric ophthalmologist at our center for 18 months, during which time the clinical findings remained stable, and no complications have been observed.

## 3. Discussion

The surgical strategy to achieve 360° access to the orbit is not standardized and varies from surgeon to surgeon. Anterior orbital tumors in the inferior, superior, and superolateral quadrants can be approached using a transconjunctival approach. For anterior orbital tumors in the inferior and/or medial quadrants, an endonasal approach, which is popular among otolaryngologists, may be appropriate [7,10]. Otherwise, a retro- or transcaruncular approach is another minimally invasive option for the medial orbit, similar to that used for medial orbital wall fractures [11]. In addition to supraorbitotomy and lateral microorbitotomy approaches [7], an upper blepharoplasty approach to the orbital roof and/or the anterior superolateral quadrants may be chosen [12]. Lesions of the superior and superolateral quadrants can be approached via a supraorbital approach [7].

However, lesions involving more than one quadrant (superior, lateral, and/or inferior) can be approached via fronto-orbital craniotomy [10], and depending on the inferior spread of the lesion, additional zygomatic osteotomy may be required [7]. Different approaches using modifications of the craniotomy and orbitotomy bone flaps have been described [13]. Unsurprisingly, complex approaches using multiple bone flaps are technically challenging, require longer operative and anesthetic times, and are more frequently associated with healing and cosmetic complications.

A simplified version of the fronto-orbital craniotomy, known as the mOZ craniotomy, was originally introduced by Balasingam et al. in neurosurgery and has been performed in pediatric patients as well [10,14,15]. mOZ craniotomy has been proven to be safe and effective in the pediatric population. Advantages include (1) ease of use; (2) superior exposure and therefore less brain retraction; (3) an easily replaceable one-piece bone flap, which obviates the need for plate sutures at the orbital rim; (4) a vascularized bone flap less susceptible to infection; and (5) the maintenance of normal temporalis muscle anatomy for improved cosmesis and function [15]. However, most publications discuss mOZ in the context of intracranial processes, and mOZ craniotomy has rarely been performed to access orbital lesions, particularly in the pediatric population.

The osteotomy technique employed in this case was performed using standard neurosurgical instruments, including a craniotome and diamond drill (Figure 4). However, the potential advantages of piezoelectric surgery, including bloodlessness, heatlessness, precision, and maneuverability, may be considered when performing mOZ.

In our case, the incomplete removal of the cyst, probably due to insufficient access via the transconjunctival route during the first operation, led to a recurrence with inflammatory manifestations. However, for the localization of the cyst residue in our case, it would not be possible to reach through an anterior orbital approach. Regarding the optimal approach in our patient, we considered the less invasive supraorbital craniotomy that can be used to access superior and superolateral orbital lesions. However, considering the complex pathology to be dealt with (residual tumor, persistent fistula, and chronic inflammatory changes (Figure 3D–F), we selected mOZ craniotomy, which allows for a wider and more direct exposure of the operative field compared with the supraorbital craniotomy, especially in the vertical axis [16]. We also excised the mass totally and sent it for pathologic consultation. It was reported as an epidermoid cyst. In our case, the benefits of this operational approach have been demonstrated to be worthwhile. The method was effective and safe, as evidenced by an acceptable functional and cosmetic result despite transient postoperative facial nerve dysfunction as a possible postoperative complication. 

To the best of our knowledge, this is the first case report of the use of mOZ craniotomy to access a residual orbital tumor in a pediatric patient. This case report is a starting point for the further study of the use of mOZ in children; due to the nature (effectiveness and safety) of mOZ, we plan to use it also in other indications in children (intraorbital tumors, bone tumors, trauma, etc.), and the results will be documented.

## 4. Conclusions

We successfully applied the mOZ craniotomy approach to access a residual superolateral orbital tumor in a pediatric patient. An mOZ one-piece craniotomy provided safe and effective exposure of the superolateral orbit with minimal brain and ocular retraction and good visual, neurological, and cosmetic outcomes.

## Figures and Tables

**Figure 1 medicina-60-01267-f001:**
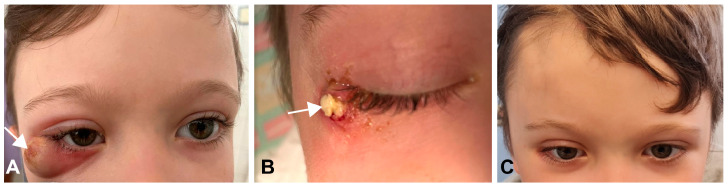
After the first (transconjunctival) surgery, a fistula and intermittent inflammatory symptoms developed in the orbital area and skin around the right outer canthus ((**A**,**B**); white arrow). After reoperation using mOZ craniotomy, good visual, neurological, and cosmetic outcomes were achieved (**C**). mOZ, modified orbitozygomatic.

**Figure 2 medicina-60-01267-f002:**
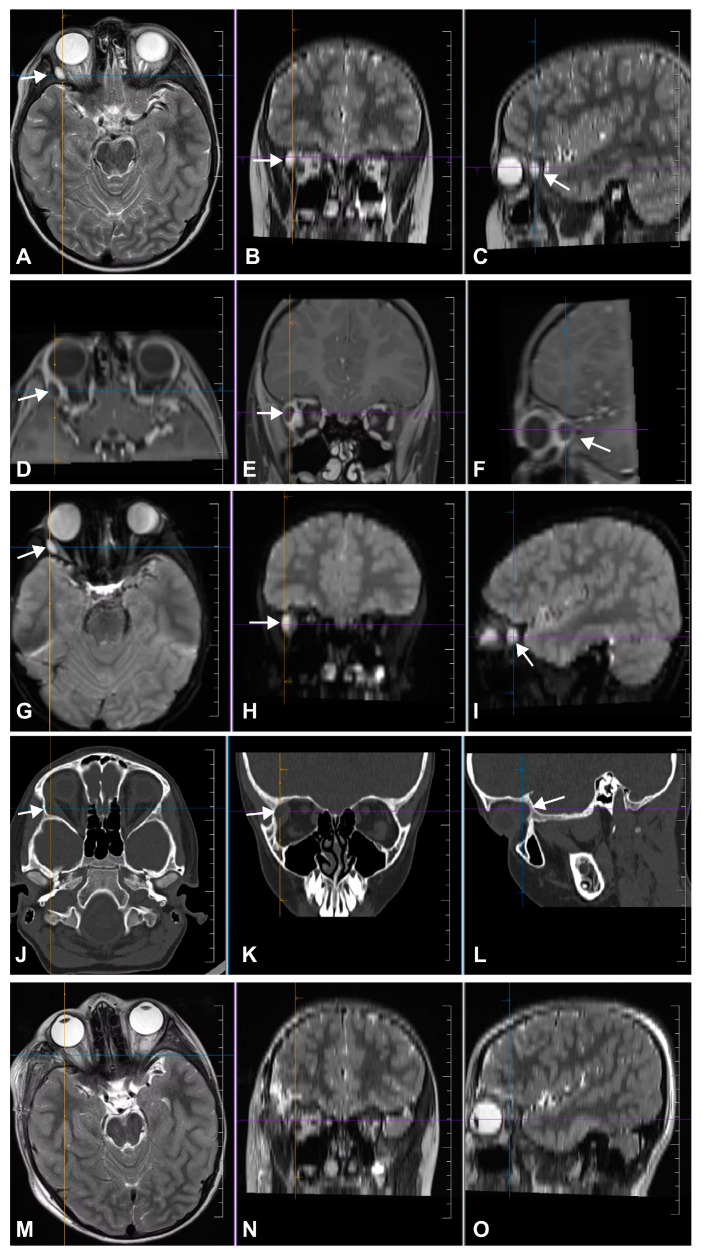
Radiographic work up. T2-(**A**–**C**) and T1-weighted MR images with contrast material (**D**–**F**) show a hyperintense lesion and an isointense lesion with postcontrast peripheral enhancement in the right superolateral orbit, respectively. The characteristic hyperintense pattern on diffusion-weighed MR images (**G**–**I**) is consistent with the diagnosis of a residual epidermoid tumor. In na-vigational CCT (**J**–**L**), an osseous indentation is observed in the area of the lesion. Postoperative T2-weighted MR images (**M**–**O**) show no residual tumor. MR, magnetic resonance; CCT, cranial computed tomography.

**Figure 3 medicina-60-01267-f003:**
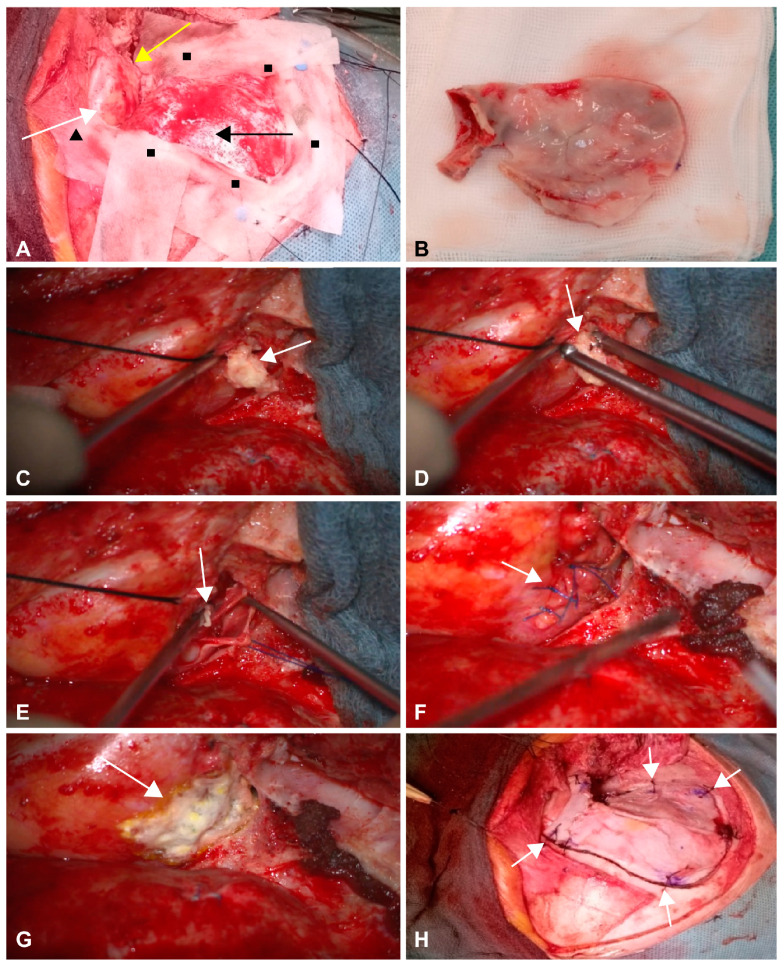
Intraoperative photographs from the reoperation using mOZ one-piece craniotomy. The patient is in supine position. The skull is rotated 45° to the left (**A**). After right-sided one-piece orbitozygomatic craniotomy, the right temporal dura mater (black arrow), right bulbus (white arrow), and orbital roof (yellow arrow) are exposed. The frontal dura mater (black triangle) and craniotomy borders (black squares) are covered with cotton patties. A one-piece orbitozygomatic bone flap is elevated, shown on a sterile gauze for demonstration (**B**) and stored in NaCl solution during the procedure. After a microsurgical incision of the periorbital region, a pearl-like tumor mass typical of an epidermoid tumor is identified ((**C**); arrow) and removed ((**D**); arrow) along with the tumor capsule ((**E**); arrow). The periorbital region is sutured using non-resorbable sutures ((**F**); arrow) and covered with a fibrin sealant patch (**G**). The bone flap is fixed to the craniotomy border using four resorbable sutures through preformed microdrill holes ((**H**); arrows). mOZ, modified orbitozygomatic.

**Figure 4 medicina-60-01267-f004:**
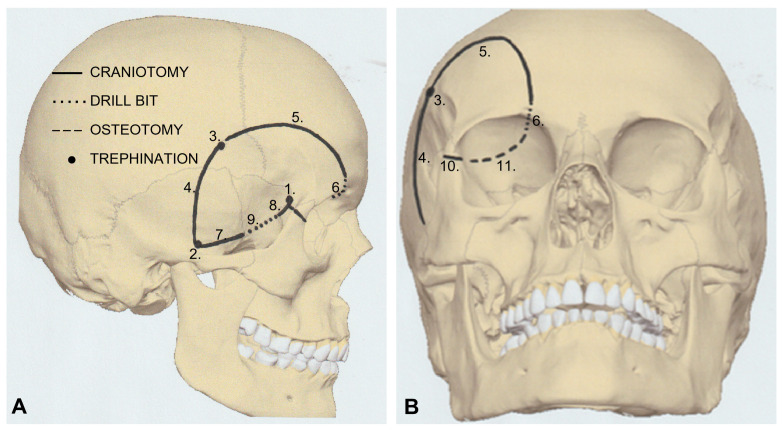
Bone work, which is a crucial part of the modified orbitozygomatic one-piece craniotomy, is demonstrated in the lateral (**A**) and anteroposterior (**B**) views. The trephinations corresponding to steps 1, 2, and 3 are performed using a diamond drill. The craniotomies corresponding to steps 4, 5, 7, 8, and 10 are performed using a standard craniotome. Steps 6, 8 and 9 are performed using a drill stripped of a footplate. Finally, the osteotomy of the orbital roof as shown in Step 11 is performed using a thin osteotome through trephination No. 1, which is placed directly over the orbital and intracranial compartments (Illustration: Robert Chrenko).

**Figure 5 medicina-60-01267-f005:**
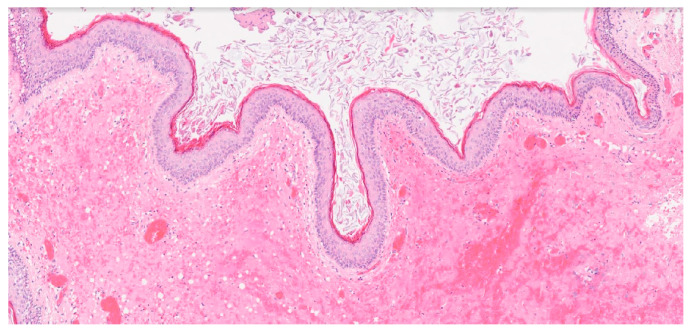
Histopathological examination shows a fibrous capsule lined with squamous epithelium, keratin, and inflammatory cells. No dysplastic transformation is detected. The patient was diagnosed with an epidermoid cyst (100×, Hematoxylin and eosin).

## Data Availability

The raw data supporting the conclusions of this article will be made available by the authors upon request.

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
