# Peer review of "Modified Orbitozygomatic Craniotomy Approach for a Recurrent Orbital Tumor in a Pediatric Patient"

_medicina, 2024, doi:10.3390/medicina60081267_

Round 1

Reviewer 1 Report

Comments and Suggestions for Authors

The case report titled "Modified orbitozygomatic craniotomy approach for a recurrent orbital tumor in a pediatric patient" presents a compelling and well-documented instance of utilizing the modified orbitozygomatic (mOZ) craniotomy to achieve complete removal of a recurrent orbital epidermoid cyst in an 8-year-old patient. The authors provide a clear rationale for selecting this approach, detailing the advantages in terms of exposure and minimizing surgical complications. The postoperative outcomes, including satisfactory visual, neurological, and cosmetic results, reinforce the efficacy and safety of this technique in pediatric patients. The report is thoroughly supported by clinical evidence and detailed surgical steps, making it a valuable contribution to pediatric neurosurgery literature.

Author Response

Comments 1: The case report titled "Modified orbitozygomatic craniotomy approach for a recurrent orbital tumor in a pediatric patient" presents a compelling and well-documented instance of utilizing the modified orbitozygomatic (mOZ) craniotomy to achieve complete removal of a recurrent orbital epidermoid cyst in an 8-year-old patient. The authors provide a clear rationale for selecting this approach, detailing the advantages in terms of exposure and minimizing surgical complications. The postoperative outcomes, including satisfactory visual, neurological, and cosmetic results, reinforce the efficacy and safety of this technique in pediatric patients. The report is thoroughly supported by clinical evidence and detailed surgical steps, making it a valuable contribution to pediatric neurosurgery literature.

Response 1: Thank you for your positive feedback.

Reviewer 2 Report

Comments and Suggestions for Authors

The clinical case discussed in the manuscript is truly impressive. Complete regression of neurological manifestations in the postoperative period indicates the correct surgical approach and the skill of the surgeons. However, there are certain comments regarding the structure of the manuscript. So in the “introduction” section, the authors should talk in more detail about the problem under consideration, the percentage prevalence in the world, especially in pediatric practice. Indicate the most common complications from typical neurosurgical approaches. In particular, lines 164-173 in the discussion make sense to move to the “introduction” section since they are devoted to general data about the neoplasm. At the same time, in the “discussion” section, it makes sense to disclose the results obtained in more detail, identifying the pros and cons of the access in question, and comparing these data with previously known data. Also in the list of references, it makes sense to include more modern sources (over the last 5 years) concerning this problem.

Author Response

Comments 1: The clinical case discussed in the manuscript is truly impressive. Complete regression of neurological manifestations in the postoperative period indicates the correct surgical approach and the skill of the surgeons. However, there are certain comments regarding the structure of the manuscript. So in the “introduction” section, the authors should talk in more detail about the problem under consideration, the percentage prevalence in the world, especially in pediatric practice. Indicate the most common complications from typical neurosurgical approaches. In particular, lines 164-173 in the discussion make sense to move to the “introduction” section since they are devoted to general data about the neoplasm. At the same time, in the “discussion” section, it makes sense to disclose the results obtained in more detail, identifying the pros and cons of the access in question, and comparing these data with previously known data. Also in the list of references, it makes sense to include more modern sources (over the last 5 years) concerning this problem.

Response 1: Thanks for pointing this out. We have made a change: lines 164-173 in the discussion have been moved to the introduction section.  In the discussion, we present that the case report is a starting point for further study of the use of mOZ in children; due to the nature (efficacy and safety) of mOZ, we plan to use it in other indications in children (intraorbital tumors, bone tumors, trauma...) and the results will be documented. 

Response 2. We would like to thank the reviewers for their valuable suggestions, which we have incorporated into the manuscript.
1. In the section entitled "Introduction", we provided a more detailed account of the issue under study, together with data on the percentage prevalence worldwide, with a particular focus on paediatric practice. 
2. A list of the most common complications resulting from typical neurosurgical approaches was provided.
3. The text from lines 164-173 has been moved from the Discussion section to the Introduction.
4. In the Discussion section, the advantages and disadvantages of the approach are presented. It should be noted that the surgical approach has not been used in children with orbital disease, and no references could be found in the literature.
5. The list of literature has been updated to include more recent sources (within the last five years) related to this issue.